# REPRESENTATION DISTILLATION BY PROTOTYPICAL CONTRASTIVE PREDICTIVE CODING

**Kyungmin Lee**
Agency for Defense Development
kyungmnlee@gmail.com

## ABSTRACT

Transferring representational knowledge of a model to another is a wide-ranging topic in machine learning. Those applications include the distillation of a large supervised or self-supervised teacher model to a smaller student model or self-supervised learning via self-distillation. Knowledge distillation is an original method to solve these problems, which minimizes a cross-entropy loss between the prototypical probabilistic outputs of teacher and student networks. On the other hand, contrastive learning has shown its competency in transferring representations as they allow students to capture the information of teacher representations. In this paper, we amalgamate the advantages of knowledge distillation and contrastive learning by modeling the critic of a contrastive objective by the prototypical probabilistic discrepancy between two features. We refer to it as *prototypical contrastive predictive coding* and present efficient implementation using the proposed objective for three distillation tasks: supervised model compression, self-supervised model compression, and self-supervised learning via self-distillation. Through extensive experiments, we validate the effectiveness of our method compared to various supervised and self-supervised knowledge distillation baselines.

## 1 INTRODUCTION

In machine learning, knowledge distillation (KD) is a problem that aims to transfer the knowledge from one network (a teacher) to another one (a student). The original method introduced by Hinton et al. (2015); Buciluă et al. (2006) minimizes the cross-entropy between the probabilistic outputs of teacher and student networks. Even though the simplicity of its implementation, many other distillation methods cannot easily outperform KD. Therefore the concept of knowledge distillation has been expanded to various machine learning tasks other than supervised model compressions, such as self-supervised learning (Caron et al., 2020; 2021) or self-supervised model compression (Fang et al., 2021). Those methods rely on minimizing the discrepancy of probabilistic outputs of teacher and student networks, thus require a function that maps a data or feature into a probability space. It is straightforward to use a linear mapping for it, which we refer to a *prototypes*.

On the other hand, contrastive learning has its merits in capturing correlations and high-order dependencies over teacher representations as they are trained by pulling the positives and pushing over the negatives. Especially, contrastive objectives such as NCE (Gutmann & Hyvärinen, 2010) or CPC (as known as InfoNCE) (Oord et al., 2018; Bachman et al., 2019; Hjelm et al., 2018) are proven to be a lower bound to the mutual information allowing student network to capture the *information* from the teacher representations. While many methods (Chen et al., 2020b;c) used those objectives for representation learning, Tian et al. (2019) proposed contrastive representation distillation (CRD) and demonstrated the effectiveness of contrastive learning in transferring knowledge from one network to another.

However, the current contrastive learning requires large negative samples with careful sampling. To ameliorate, we combine the prototypical method and contrastive objective to inherit their advantages. To that end, we propose *prototypical contrastive predictive coding* (ProtoCPC), which utilizes the prototypes to generate probability distribution and model the critic by the discrepancy between two probabilistic outputs. Furthermore, our ProtoCPC objective is a lower bound to the mutual

information between teacher and student representations alike CPC. But unlike CPC, it does not require commodious negative samples or careful sampling policies.

Given the ProtoCPC objective, we present efficient applications on three distillation tasks: supervised model compression, self-supervised model compression, and self-supervised learning via self-distillation. While ProtoCPC is simple in its implementation, we observe that it significantly boosts the performance of representation distillation. Our method outperforms KD, and partially outperform CRD on CIFAR-100 and ImageNet. Moreover, our method achieves state-of-the-art performance on transferring various self-supervised teacher representations to a small self-supervised model. Lastly, by construing self-supervised learning as a self-distillation, we validate the effectiveness of ProtoCPC in representation learning. Our contributions are following:

- We propose a novel prototypical contrastive objective for transferring representational knowledge between models.
- We apply the proposed objective to three distillation tasks: supervised model compression, self-supervised model compression, and self-supervised learning via self-distillation.
- Experiments show the effectiveness of our method in various representation distillation benchmarks, especially achieving state-of-the-art performance in supervised/self-supervised model compression.

## 2 METHOD

### 2.1 PROTOTYPICAL CONTRASTIVE PREDICTIVE CODING

Given data $x$ with a random variable $x \sim X$, let the teacher network $f^T$ and the student network $f^S$, where they map $x$ into $\mathbb{R}^{D_T}$ and $\mathbb{R}^{D_S}$ respectively. Also let $T$ and $S$ be random variables for representation $f^T(X)$ and $f^S(X)$ respectively. One can transfer the representational knowledge of $T$ to $S$ by maximizing the mutual information $I(T;S)$, where it is defined by the KL-divergence between the joint distribution $p(T,S)$ and the product of marginal distribution $p(T)p(S)$. However, as estimation and optimization of mutual information is challenging, many approaches count on maximizing variational lower bound to the mutual information (Poole et al., 2019). The contrastive predictive coding (Oord et al., 2018), or as known as InfoNCE is a guaranteed lower bound to the mutual information (Oord et al., 2018; Tian et al., 2020; Bachman et al., 2019; Song & Ermon, 2020), and has shown its competency in both representation learning (He et al., 2020; Chen et al., 2020b;c) and representation distillation (Tian et al., 2019). Formally, given a $z_s \sim S$ with a positive $z_t \equiv z_{t0}$ and $N-1$ negatives $\{z_{tj}\}_{j=1}^{N-1}$ sampled from $T$, i.e. $(z_t, z_s) \sim p(T,S)$ and $\{(z_{tj}, z_s)\}_{j=1}^{N-1} \sim p(T)p(S)$, the following inequality holds for any critic $h : \mathbb{R}^{D_T} \times \mathbb{R}^{D_S} \to \mathbb{R}_+$:

$$I(T;S) \geq \mathbb{E}\left[ \log \frac{h(z_t, z_s)}{\frac{1}{N}\sum_{j=0}^{N-1} h(z_{tj}, z_s)} \right] \tag{1}$$

Previous works set the critic by the exponential of cosine similarity between two unit feature vectors, i.e. $h(z_t, z_s) = \exp(-z_t \cdot z_s)$ where $z_t$ and $z_s$ are $\ell_2$-normalized. Theoretically, the lower bound becomes tighter as $N \to \infty$. In practice, the CPC objective requires using extremely large batch size or memory buffer that stores the negatives as it requires pairwise computation between $z_s$ and $z_{tj}$s.

On the other hand, we project feature vectors into a probability space. To do that, we append a linear prototypes $W^T \in \mathbb{R}^{D_T \times K}$ and $W^S \in \mathbb{R}^{D_S \times K}$ at the top of $f^T$ and $f^S$ so that they have same output dimension of $K$. For brevity, let $\bar{z}_s = W^S z_s$ and $\bar{z}_t = W^T z_t$ and $\bar{T}, \bar{S}$ be random variables for $\bar{z}_t$ and $\bar{z}_s$ respectively. Then we set probability of student $p_s$ by $K$-categorical distribution defined by the softmax operator on $\bar{z}_s = W^S z_s$ with temperature $\tau_s > 0$:

$$p_s^{(k)} = \frac{\exp\left(\bar{z}_s^{(k)}/\tau_s\right)}{\sum_{k'=1}^{K} \exp\left(\bar{z}_s^{(k')}/\tau_s\right)}. \tag{2}$$

Similarly we define probability of teacher $p_{tj}$ with temperature $\tau_t > 0$. Then we define the critic between $\bar{z}_t$ and $\bar{z}_s$ by the negative exponential of cross-entropy between $p_s$ and $p_t$, i.e.,

$$h(\bar{z}_t, \bar{z}_s) = e^{-H(p_t, p_s)} = e^{\sum_{k=1}^{K} p_t^{(k)} \log p_s^{(k)}}. \tag{3}$$

Then $h$ is a positive bounded function and is maximized when $p_t$ matches with $p_s$. Then by plugging Eq. 3 into Eq. 1, it follows that

$$I(T; S) \geq I(\bar{T}; \bar{S}) \geq \mathbb{E}\left[\log \frac{e^{-H(p_t, p_s)}}{\frac{1}{N}\sum_{j=0}^{N-1} e^{-H(p_{tj}, p_s)}}\right] \quad (4)$$

$$= \mathbb{E}\left[\log \frac{\exp(p_t \cdot \bar{z}_s / \tau_s)}{\frac{1}{N}\sum_{j=0}^{N-1} \exp\left(p_{tj} \cdot \bar{z}_s / \tau_s\right)}\right] \quad (5)$$

$$\geq \mathbb{E}\left[\log \frac{\exp(p_t \cdot \bar{z}_s / \tau_s)}{\frac{1}{N}\sum_{j=0}^{N-1}\sum_{k=1}^{K} p_{tj}^{(k)} \exp\left(\bar{z}_s^{(k)} / \tau_s\right)}\right] \quad (6)$$

$$= \mathbb{E}\left[\log \frac{\exp(p_t \cdot \bar{z}_s / \tau_s)}{\sum_{k=1}^{K} q^{(k)} \exp\left(\bar{z}_s^{(k)} / \tau_s\right)}\right] \triangleq I_{\text{ProtoCPC}}, \quad (7)$$

where $q^{(k)} = \frac{1}{N}\sum_{j=0}^{N-1} p_{tj}$ is a mean of teachers' probability which we call a *prior*. The first inequality is from data processing inequality, the second equality is from crossing out the constant term, and third inequality is from Jensen's inequality. We define **Prototypical Contrastive Predictive Coding (ProtoCPC)** objective $I_{\text{ProtoCPC}}$ in Eq. 7. In addition, we define **ProtoCPC loss** $\mathcal{L}_{\text{ProtoCPC}}$ by the negative of ProtoCPC objective, thus minimizing ProtoCPC loss is equivalent to a variational maximization of mutual information between student and teacher representations.

**Relationship with CPC**   While many lower bounds to the mutual information were proposed, Tschannen et al. (2019) observe that the tightness of bound does not necessarily imply a better representation learning performance. From then, many works focused on analyzing the components of contrastive objective itself which are responsible for the empirical success. Wang & Isola (2020) argued that the contrastive loss is composed of alignment and uniformity loss, where alignment loss accounts for the similarity of two positive features, and uniformity loss measures how the features are scattered in the unit hypersphere and show that both losses are important in contrastive learning.

We draw an analogy on ProtoCPC by dissecting into alignment and uniformity losses. Since the alignment loss is straightforward, we focus on the uniformity loss. We show that one can interpret the uniformity loss $\mathcal{L}_{\text{ProtoCPC-Unif}}$ by the re-substitution entropy estimator of $z_s$ via a von-Mises Fisher kernel density estimation (vMF-KDE) (Ahmad & Lin, 1976):

$$\mathcal{L}_{\text{ProtoCPC-Unif}} = \mathbb{E}_{z_s \sim S}\left[\log \sum_{k=1}^{K} q^{(k)} \exp(\bar{z}_s^{(k)} / \tau_s)\right] = \mathbb{E}_{z_s \sim S}\left[\log \sum_{k=1}^{K} q^{(k)} \exp(w_k \cdot z_s / \tau_s)\right] \quad (8)$$

$$= \mathbb{E}_{z_s \sim S}[\log \hat{p}_{\text{vMF-KDE}}(z_s)] + \log Z_{\text{vMF}} = -\hat{H}(z_s) + \log Z_{\text{vMF}}, \quad (9)$$

where each $w_k$ is a $k$-th column of $W_S$ and acts as a mean direction of $k$-th vMF distribution and $q^{(k)}$ acts as a prior for each $k$-th vMF distribution. The $\hat{p}_{\text{vMF-KDE}}$ is thus the mixture of $K$ vMF distribution with prior $q^{(k)}$ and then the uniformity loss is a re-substitution entropy $\hat{H}(z_s)$. The $Z_{\text{vMF}}$ is a normalizing constant for vMF distribution. Remark that the uniformity loss of CPC objective is also a re-substitution entropy with vMF-KDE, but the mean directions are given by negative samples $z_{tj}$ and the prior is uniform. It shows that the ProtoCPC objective allows modeling of complex mixture of vMF distribution by exploiting prior term and using prototypes remove the dependency on negative samples.

**Prior momentum**   Since ProtoCPC is contrastive, it requires sufficient negatives to perform learning. However, unlike CPC, ProtoCPC only requires prior $q$ that accounts for the negatives. Therefore, we use exponential moving average (EMA) on prior $q$ for better estimation. At each iteration, we update $q$ by following update rule:

$$q^{(k)} \leftarrow m_p q^{(k)} + (1 - m_p)\frac{1}{N}\sum_{j=1}^{N} p_{tj}^{(k)}, \quad (10)$$

where $m_p > 0$ is a momentum rate. The prior momentum allows better estimation of prior term regardless of the size of negative samples.

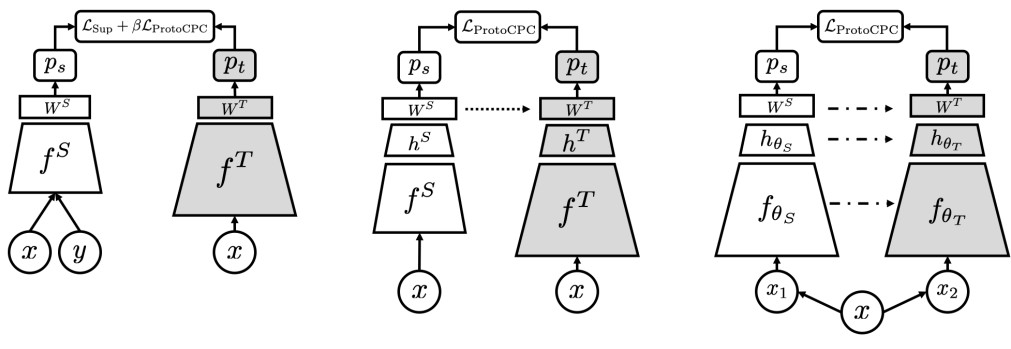

(a) Supervised Model Compression   (b) Self-supervised Model Compression   (c) Self-distillation

Figure 1: Demonstration of three distillation tasks using proposed ProtoCPC objective. (a) Compressing a supervised model, (b) compressing a self-supervised model, (c) self-supervised learning via self-distillation. Our method is a contrative objective with the probability distributions where it is mediated by the prototypes. The figures of white color is training parameters while gray colors are frozen throughout the training. The dashed lines represent the copying of parameters such as exponential moving average (EMA).

**Assignment of teacher probability**   While KD used softmax operator for both probabilities of teacher and student networks, many self-supervised methods (Asano et al., 2019; Caron et al., 2020) reported that the softmax operator can lead to collapse, i.e. every representation fall into the same one. To compromise, many prototypical self-supervised methods resort on *Sinkhorn-Knopp* iterative algorithm by formulating the assignment of teacher probability as an optimal transport problem:

$$\underset{P_t}{\arg\max} \langle P_t, Z_t \rangle + \tau_t H(P_t), \quad \text{s.t.} \quad P_t \in \mathbb{R}_+^{N \times K}, P_t \mathbf{1}_K = \mathbf{1}_N, P_t^\top \mathbf{1}_N = \frac{N}{K} \mathbf{1}_K, \quad (11)$$

where $Z_t$ is a matrix whose rows are $z_{tj}$, $H(P_t) = \sum_{j=0}^{N-1} \sum_{k=1}^{K} -p_t^{(k)} \log p_t^{(k)}$ is a entropy and $\tau_t > 0$ is a temperature that controls the smoothness of distribution. Then the Eq. 11 can be solved by only few steps of Sinkhorn-Knopp iteration (Cuturi, 2013; Altschuler et al., 2017) which iteratively projects $P_t$ into following form:

$$p_{tj}^{(k)} = \frac{\beta_k e^{z_{tj}^{(k)}/\tau_t}}{\sum_{k'=1}^{K} \beta_{k'} e^{z_{tj}^{(k')}/\tau_t}}, \quad (12)$$

where $\beta_k$ is a normalizing constant. We refer this to Sinkhorn-Knopp (SK) operator. The SK operator allows each prototype to be non-zero, i.e. $q^{(k)}$ is non-zero for each $k$. In practice, we only conduct a few steps of SK iterations.

## 2.2 Applications of ProtoCPC to distillation tasks

Fig. 1 depicts explanation on how we structure prototypical contrastive learning for three distillation tasks we consider: supervised model compression, self-supervised model compression and self-distillation.

**Supervised model compression**   Given a (supervised) pre-trained teacher network $f^T$, the goal of supervised model compression is to train a smaller student network $f^S$ by utilizing the representational knowledge from the teacher. To do that, Hinton et al. (2015) proposed to minimize the knowledge distillation loss, where it is a weighted sum of the supervised loss and the distillation loss. Here, the supervised loss is a conventional cross-entropy loss between the one-hot label and the output of $f^S$. The distillation loss is a cross-entropy between teacher's probability output $p_t$ and student's probability output $p_s$. Formally, it is given by following:

$$\min_{f^S} \mathbb{E}_{(x,y) \sim (X,Y)} \big[ \mathcal{L}_{\text{Sup}}(f^S(x), y) + \beta \tau^2 H(p_t(x), p_s(x)) \big], \quad (13)$$

where $y$ is a label, $p_t(x)$ and $p_s(x)$ are computed by the softmax operator on each $f^T(x)$ and $f^S(x)$ with temperature $\tau > 0$, and $\beta$ is a balancing weight. Our approach is to replace the distillation loss by the ProtoCPC loss and use SK operator for $p_t$. Then our loss becomes following:

$$\min_{f^S} \mathbb{E}_{(x,y) \sim (X,Y)} \big[ \mathcal{L}_{\text{Sup}}(f^S(x), y) + \beta \tau^2 \mathcal{L}_{\text{ProtoCPC}}(p_t(x), p_s(x)) \big]. \quad (14)$$

Remark that since the supervised networks already contain prototypes at the last layer, we use prototypes for both supervised learning and distillation at the same time. Note that CRD (Tian et al., 2019) uses binary contrastive loss for distillation loss by maintaining a large memory buffer of negative samples. Also, CRD requires an additional linear layer to implement inner-product critic. However, our method conducts contrastive learning without additional training of embeddings or storing a memory buffer of negative samples.

**Self-supervised model compression**   Given a large self-supervised pre-trained teacher network $g^T$, the goal of self-supervised model compression is to transfer the representational knowledge of $g^T$ to a smaller self-supervised student network. Fang et al. (2021) first proposed SEED, where the student network is trained by minimizing the cross-entropy loss between the similarity score of teacher and student. The similarity score is computed by the pairwise computation over a memory buffer filled with features of the teacher network.

Our approach uses prototypes to generate the probabilistic output of teacher and student representations, and use ProtoCPC loss for effective distillation. Let the pre-trained teacher network $g^T$ be a composition of base encoder $f^T$ and the projection head $h^T$. Then we train student network $g^S$, where it is composed of smaller encoder $f^S$ and projection head $h^S$ of the same architecture as $h^T$. Then we append prototypes $W^T$ and $W^S$ for each $g^T$ and $g^S$ to ensure that they have the same output of dimension $K$. Given a data $x$, we train the student network by minimizing the ProtoCPC loss between the probability of teacher $p_t(x)$ and probability of student $p_s(x)$. For $p_s$ we use softmax operator with temperature $\tau_s > 0$ and for $p_t$ we use SK operator with temperature $\tau_t > 0$ over the batch of samples. Then the objective is given by following:

$$\min_{g^S, W^S} \mathbb{E}_{x \sim X} \left[ \mathcal{L}_{\text{ProtoCPC}}(p_t(x), p_s(x)) \right]. \tag{15}$$

We can further use multi-crops data augmentation (Caron et al., 2020), where it augments multiple small crops of a data during the training to expedite the training. Let $\tilde{x}_j \sim \mathcal{M}(x)$ be small local crops with transformation $\mathcal{M}$, then the self-supervised model compression with multi-crops is given by following:

$$\min_{g^S, W^S} \mathbb{E}_{x \sim X, \tilde{x}_j \sim \mathcal{A}(x)} \left[ \mathcal{L}_{\text{ProtoCPC}}(p_t(x), p_s(x)) + \sum_j \mathcal{L}_{\text{ProtoCPC}}(p_t(x), p_s(\tilde{x}_j)) \right]. \tag{16}$$

For prototypes of teacher network $W^T$, we copy the parameters of $W^S$ to $W^T$ at each iteration. This allows our method to apply to any self-supervised teacher networks. Note that if the teacher network is trained by prototypical methods such as DINO (Caron et al., 2021) or SwAV (Caron et al., 2020), we can re-use the pre-trained prototypes for $W^T$. We present ablation studies on setting prototypes for $W^T$.

**Self-distillation**   Additionally, we show that ProtoCPC loss can enhance the representation learning of a self-supervised model, where we interpret it as a self-distillation method. We built on DINO (Caron et al., 2021), which is the state-of-the-art method in self-supervised learning that first regarded the self-supervised learning as a knowledge distillation problem.

Let the student network be the composition of a base encoder $f_{\theta_S}$, we append projection head $h_{\theta_S}$ consists of MLP layers and a prototypical layer $W_{\theta_S}$. Then we set the teacher by the momentum encoder (He et al., 2020; Chen et al., 2020b; Grill et al., 2020; Caron et al., 2021) on the student network, where it is a composition of encoder $f_{\theta_T}$, projection head $h_{\theta_T}$, and prototype $W_{\theta_T}$. We use EMA for the weights of teacher network, where it is update by $\theta_t \to \lambda \theta_t + (1 - \lambda)\theta_s$, where $\lambda$ is a momentum rate. Within the training, we generate a pair of views $(x_1, x_2)$ from a data $x \sim X$ and they are passed to each student and teacher networks. Then we compute ProtoCPC loss between the probability of teacher network $p_t(x_1)$ and probability of student network $p_s(x_2)$, and symmetrically between $p_t(x_2)$ and $p_s(x_1)$. The objectives are given by:

$$\min_{\theta_s} \mathbb{E}_{x \sim X, x_1, x_2 \sim \mathcal{A}(x)} \left[ \frac{1}{2} \mathcal{L}_{\text{ProtoCPC}}(p_t(x_1), p_s(x_2)) + \frac{1}{2} \mathcal{L}_{\text{ProtoCPC}}(p_t(x_2), p_s(x_1)) \right], \tag{17}$$

where $\mathcal{A}(\cdot)$ is a data augmentation operator. Note that multi-crops data augmentation can be further applied.

## 3   RELATED WORK

**Contrastive learning**   There is a large body of works on contrastive objectives for representation learning. The contrastive objectives such as CPC (Oord et al., 2018; Song & Ermon, 2020; Bachman et al., 2019; Hjelm et al., 2018) and multi-view data augmentation (Tian et al., 2020) are keys for the success of contrastive representation learning. With sophisticated implementation such as momentum encoder (He et al., 2020; Chen et al., 2020b; Grill et al., 2020), or caching negative samples (Wu et al., 2018; Chen et al., 2020b;c) had dramatically reduced the gap between self-supervised and supervised models. While the contrastive objectives are proven to be a lower bound to the mutual information (Poole et al., 2019; Song & Ermon, 2020), some studies attribute that the success of representation learning to the contrastive objective itself rather than the tightness of mutual information estimation (Bachman et al., 2019). Then many works focus on analyzing the contrastive objective itself (Wang & Isola, 2020) or count on the metric-learning perspective of contrastive learning that deviates into different objectives (Grill et al., 2020; Chen & He, 2021; Zbontar et al., 2021; Bardes et al., 2021). The key difference between many contrastive objectives and our method is that their discrepancy measure is given by the $\ell_2$-norm of two features, while our method projects feature into a probability space and characterize the discrepancy with probabilistic measure. Our work is also related to methods that use clustering for representation learning (Tian et al., 2017; Zhuang et al., 2019; Li et al., 2020), which have been developed into prototypical methods (Asano et al., 2019; Li et al., 2020; Caron et al., 2020; 2021).

**Knowledge distillation**   The idea that transferring the large pre-trained model's representation into a smaller one has been embodied by knowledge distillation (Hinton et al., 2015; Buciluǎ et al., 2006). Then various distillation criteria or utilization of intermediate feature maps were studied to enhance the performance (Romero et al., 2014; Zagoruyko & Komodakis, 2016a; Tung & Mori, 2019; Peng et al., 2019; Ahn et al., 2019; Park et al., 2019; Passalis & Tefas, 2018; Heo et al., 2019; Kim et al., 2018; Yim et al., 2017; Huang & Wang, 2017). However, many of them were not able to outperform KD by far. Recently, Tian et al. (2019) introduced *contrastive representation distillation* (CRD) which utilizes binary contrastive objective for distillation loss and shows the empirical superiority on various knowledge distillation benchmarks.

Moreover, recent studies focus on distilling the knowledge of large self-supervised models to the smaller ones using the concept of knowledge distillation (Fang et al., 2021; Shen et al., 2021; Noroozi et al., 2018; Chen et al., 2020a). Fang et al. (2021) proposed self-supervised representation distillation (SEED) which used features of teacher networks to compute probability distribution for distillation. One can interpret SEED as the prototypical method where prototypes are given by the queue of teacher features.

## 4   EXPERIMENT

### 4.1   SUPERVISED MODEL COMPRESSION

**Setup**   We experiment on CIFAR-100 (Krizhevsky et al., 2009) and ImageNet (Deng et al., 2009) with various teacher-student combinations such as ResNet (He et al., 2016) and Wide ResNet (WRN) (Zagoruyko & Komodakis, 2016b). We compare with various distillation baselines such as KD (Hinton et al., 2015) and CRD (Tian et al., 2019).

**Results on CIFAR-100**   Table 1 and Table 2 compare top-1 *accuracies* of our method on supervised model compression on CIFAR-100. Table 1 investigates teachers and students of the same architectural style, while Table 2 is focused on students and teachers from different architectures. We observe that our ProtoCPC consistently outperforms KD and other distillation methods, yet ProtoCPC is on par with CRD where it also uses a contrastive learning method for distillation. Remark that CRD requires a large memory buffer of negative samples with careful sampling according to the class information and additional training of embeddings. However, our ProtoCPC does not require massive negative samples and additional training of linear embeddings since we use the last linear layer as prototypes for our ProtoCPC.

| Teacher
Student | WRN-40-2
WRN-16-2 | WRN-40-2
WRN-40-1 | resnet56
resnet20 | resnet110
resnet20 | resnet110
resnet32 | resnet32x4
resnet8x4 | vgg13
vgg8 |
|---|---|---|---|---|---|---|---|
| Teacher | 75.61 | 75.61 | 72.34 | 74.31 | 74.31 | 79.42 | 74.64 |
| Student | 73.26 | 71.98 | 69.06 | 69.06 | 71.14 | 72.50 | 70.36 |
| KD | 74.92 | 73.54 | 70.66 | 70.67 | 73.08 | 73.33 | 72.98 |
| FitNet | 73.58 (↓) | 72.24 (↓) | 69.21 (↓) | 68.99 (↓) | 71.06 (↓) | 73.50 (↑) | 71.02 (↓) |
| AT | 74.08 (↓) | 72.77 (↓) | 70.55 (↓) | 70.22 (↓) | 72.31 (↓) | 73.44 (↑) | 71.43 (↓) |
| SP | 73.83 (↓) | 72.43 (↓) | 69.67 (↓) | 70.04 (↓) | 72.69 (↓) | 72.94 (↓) | 72.68 (↓) |
| CC | 73.56 (↓) | 72.21 (↓) | 69.63 (↓) | 69.48 (↓) | 71.48 (↓) | 72.97 (↓) | 70.71 (↓) |
| VID | 74.11 (↓) | 73.30 (↓) | 70.38 (↓) | 70.16 (↓) | 72.61 (↓) | 73.09 (↓) | 71.23 (↓) |
| RKD | 73.35 (↓) | 72.22 (↓) | 69.61 (↓) | 69.25 (↓) | 71.82 (↓) | 71.90 (↓) | 71.48 (↓) |
| PKT | 74.54 (↓) | 73.45 (↓) | 70.34 (↓) | 70.25 (↓) | 72.61 (↓) | 73.64 (↑) | 72.88 (↓) |
| AB | 72.50 (↓) | 72.38 (↓) | 69.47 (↓) | 69.53 (↓) | 70.98 (↓) | 73.17 (↓) | 70.94 (↓) |
| FT | 73.25 (↓) | 71.59 (↓) | 69.84 (↓) | 70.22 (↓) | 72.37 (↓) | 72.86 (↓) | 70.58 (↓) |
| FSP | 72.91 (↓) | n/a | 69.95 (↓) | 70.11 (↓) | 71.89 (↓) | 72.62 (↓) | 70.23 (↓) |
| NST | 73.68 (↓) | 72.24 (↓) | 69.60 (↓) | 69.53 (↓) | 71.96 (↓) | 73.30 (↓) | 71.53 (↓) |
| CRD | 75.48 (↑) | 74.14 (↑) | 71.16 (↑) | **71.46** (↑) | 73.48 (↑) | **75.51** (↑) | **73.94** (↑) |
| ProtoCPC | **75.79** (↑) | **74.23** (↑) | **71.41** (↑) | 71.04 (↑) | **73.55** (↑) | 75.02 (↑) | 73.79 (↑) |
| KD+CRD | 75.64 (↑) | 74.38 (↑) | 71.63 (↑) | 71.56 (↑) | 73.75 (↑) | 75.46 (↑) | 74.29 (↑) |
| ProtoCPC+CRD | 75.92 (↑) | 74.75 (↑) | 70.99 (↑) | 71.47 (↑) | 73.52 (↑) | 75.55 (↑) | 74.40 (↑) |

Table 1: Top-1 test *accuracy* (%) of student networks on CIFAR-100 with various distillation methods (our method denoted by ProtoCPC). (↑) denotes outperformance over KD and bold font denotes the best accuracy within the baseline. Note that ProtoCPC always outperform KD as well as other baselines except CRD. Our ProtoCPC outperforms CRD in 4 out of 7 benchmarks. We also provide results on combining ProtoCPC and CRD, showing that our method is compatible with CRD as they learn different structure. The results are averaged over 5 runs.

| Teacher
Student | vgg13
MobileNetV2 | ResNet50
MobileNetV2 | ResNet50
vgg8 | resnet32x4
ShuffleNetV1 | resnet32x4
ShuffleNetV2 | WRN-40-2
ShuffleNetV1 |
|---|---|---|---|---|---|---|
| Teacher | 74.64 | 79.34 | 79.34 | 79.42 | 79.42 | 75.61 |
| Student | 64.6 | 64.6 | 70.36 | 70.5 | 71.82 | 70.5 |
| KD | 67.37 | 67.35 | 73.81 | 74.07 | 74.45 | 74.83 |
| FitNet | 64.14 (↓) | 63.16 (↓) | 70.69 (↓) | 73.59 (↓) | 73.54 (↓) | 73.73 (↓) |
| AT | 59.40 (↓) | 58.58 (↓) | 71.84 (↓) | 71.73 (↓) | 72.73 (↓) | 73.32 (↓) |
| SP | 66.30 (↓) | 68.08 (↑) | 73.34 (↓) | 73.48 (↓) | 74.56 (↑) | 74.52 (↓) |
| CC | 64.86 (↓) | 65.43 (↓) | 70.25 (↓) | 71.14 (↓) | 71.29 (↓) | 71.38 (↓) |
| VID | 65.56 (↓) | 67.57 (↑) | 70.30 (↓) | 73.38 (↓) | 73.40 (↓) | 73.61 (↓) |
| RKD | 64.52 (↓) | 64.43 (↓) | 71.50 (↓) | 72.28 (↓) | 73.21 (↓) | 72.21 (↓) |
| PKT | 67.13 (↓) | 66.52 (↓) | 73.01 (↓) | 74.10 (↑) | 74.69 (↑) | 73.89 (↓) |
| AB | 66.06 (↓) | 67.20 (↓) | 70.65 (↓) | 73.55 (↓) | 74.31 (↓) | 73.34 (↓) |
| FT | 61.78 (↓) | 60.99 (↓) | 70.29 (↓) | 71.75 (↓) | 72.50 (↓) | 72.03 (↓) |
| NST | 58.16 (↓) | 64.96 (↓) | 71.28 (↓) | 74.12 (↑) | 74.68 (↑) | 74.89 (↑) |
| CRD | **69.73** (↑) | 69.11 (↑) | 74.30 (↑) | 75.11 (↑) | **75.65** (↑) | 76.05 (↑) |
| ProtoCPC | 69.09 (↑) | **69.50** (↑) | **74.32** (↑) | **75.24** (↑) | 76.50 (↑) | **76.28** (↑) |
| KD+CRD | 69.94 (↑) | 69.54 (↑) | 74.58 (↑) | 75.12 (↑) | 76.05 (↑) | 76.27 (↑) |
| ProtoCPC+CRD | 69.77 (↑) | 70.79 (↑) | 74.95 (↑) | 76.01 (↑) | 76.27 (↑) | 76.82 (↑) |

Table 2: Top-1 test *accuracy*(%) of student networks on CIFAR-100 of a various distillation methods (ours is ProtoCPC) for transfer across very different teacher and student architectures. ProtoCPC outperforms KD and all other methods except CRD. Our method outperforms CRD on 4 out of 6 benchmarks. We also observe that combining ProtoCPC and CRD boosts the performance significantly. Average over 3 runs.

**Results on ImageNet** In Table 3, we compare our ProtoCPC to different distillation methods such as AT (Zagoruyko & Komodakis, 2016a) and Online-KD (Lan et al., 2018) on ImageNet, where pre-trained ResNet-34 is used for teacher and ResNet-18 is a student. We observe that ProtoCPC outperforms various knowledge distillation baselines such as KD, and is slightly better than CRD. Remark that our method is simpler in its implementation and shows better performance than CRD.

|        | Teacher | Student | AT | KD | SP | CC | Online KD | CRD | ProtoCPC |
|--------|---------|---------|------|------|------|-------|-----------|------|----------|
| Top-1  | 26.69   | 30.25   | 29.30 | 29.34 | 29.38 | 30.04 | 29.45     | 28.83 | **28.54** |
| Top-5  | 8.58    | 10.93   | 10.00 | 10.12 | 10.20 | 10.83 | 10.41     | 9.87 | **9.49**  |

Table 3: Top-1 and Top-5 error rates (%) of student network ResNet-18 on ImageNet validation set. For fair comparison, we use same setting as in Tian et al. (2019). We compare our ProtoCPC with KD (Hinton et al., 2015), AT (Zagoruyko & Komodakis, 2016a), SP (Tung & Mori, 2019), CC (Peng et al., 2019), Online KD (Lan et al., 2018), and CRD (Tian et al., 2019).

## 4.2 SELF-SUPERVISED MODEL COMPRESSION

**Setup**  We experiment distillation of various self-supervised networks to ResNet-18 (He et al., 2016) on ImageNet (Deng et al., 2009) without class labels. We consider following self-supervised teacher networks: MoCo-v2 (Chen et al., 2020b) pre-trained ResNet-50, SwAV (Caron et al., 2020) pre-trained ResNet-50 and DINO (Caron et al., 2021) pre-trained ResNet-50 and vision transformer (Dosovitskiy et al., 2020). We train for 100 epochs and we additionally conduct experiments on using multi-crops data augmentation for SwAV and DINO pre-trained ResNet-50 networks. For evaluation, we follow linear evaluation protocol which conducts supervised learning on the linear layer appended at the top of the frozen feature and $k$-nearest neighbor classification ($k$-NN).

**Main results**  Table 4 show the main results of our self-supervised model compression compared to self-supervised learning (SSL) of itself. We observe that ProtoCPC outperforms SSL with a large margin, especially showing superior performance in $k$-NN classification. Note that our method works well for various self-supervised teacher networks and even works well when the teacher and student networks are of different architectures (vision transformer teacher to ResNet student).

|            | MoCo ResNet-50 | | SwAV ResNet-50 | | DINO ResNet-50 | | DINO DeiT-S/16 | |
|------------|--------|--------|--------|--------|--------|--------|--------|--------|
|            | *Linear* | *k-NN* | *Linear* | *k-NN* | *Linear* | *k-NN* | *Linear* | *k-NN* |
| Teacher    | 71.1   | 61.9   | 75.3   | 65.7   | 75.3   | 67.5   | 77.0   | 74.3   |
| Supervised | 69.5   | 69.5   | 69.5   | 69.5   | 69.5   | 69.5   | 69.5   | 69.5   |
| SSL        | 52.5   | 36.7   | 57.5   | 48.2   | 58.2   | 50.3   | 58.2   | 50.3   |
| ProtoCPC   | **61.1** | **55.6** | **63.1** | **57.7** | **63.5** | **60.3** | **65.5** | **63.2** |

Table 4: Main result of our ProtoCPC on distillation of various self-supervised teacher models to ResNet-18. The teacher models are MoCo (Chen et al., 2020b) ResNet-50, SwAV (Caron et al., 2020) ResNet-50 and DINO (Caron et al., 2021) ResNet-50 and DeiT small with patch size 16. The self-supervised denotes the result of self-supervised learning on ResNet-18 with the same method of teacher network.

**Comparison with SEED**  Table 5 compares our method with SEED (Fang et al., 2021), an original method for self-supervised model compression. We observe that our method consistently outperforms SEED in both $k$-NN and *linear* evaluation with the same teacher network applied. Although training for shorter epochs, our method achieves outperforms SEED. When the teacher is pre-trained by SwAV, our method outperforms SEED without using multi-crops data augmentation, and the gap becomes larger when we use multi-crops as well. When using DINO self-supervised teacher and multi-crops data augmentation, we achieve the best results in representation learning of ResNet-18.

| Teacher | Method | Epochs | *Linear* | *k-NN* |
|---------|--------|--------|----------|--------|
| MoCo    | SEED     | 200  | 60.5 | 49.1 |
|         | ProtoCPC | 100  | **61.1** | **55.6** |
| SwAV    | SEED     | 100  | 61.1 | - |
|         | SEED     | 200* | 62.6 | - |
|         | ProtoCPC | 100  | 63.1 | **57.7** |
|         | ProtoCPC | 100* | **63.9** | 57.0 |
| DINO    | ProtoCPC | 100  | 63.5 | 60.3 |
|         | ProtoCPC | 100* | **65.3** | **60.7** |

Table 5: Comparison of our method with SEED. * denotes training with multi-crops. Every teacher networks are ResNet-50 and student networks are ResNet-18.

## 4.3 SELF-DISTILLATION

**Setup**  We evaluate our ProtoCPC objective in self-supervised learning, where we cast it as a self-distillation. We follow DINO (Caron et al., 2021) for experimental procedures, where the teacher

network is set by the momentum encoder, and linear prototypes are attached to the features to distill the knowledge.

We change the loss function from cross-entropy loss to ProtoCPC loss without any adjustment in other settings. We use a prior momentum rate of 0.9 for ProtoCPC loss. We evaluate on both ResNet-50 and DeiT-S/16, where DINO performs well on both architectures. For evaluation, we report the results of linear evaluation protocol and $k$-nn classification.

**Results** Table 6 compare the performance of representation learning of our method and DINO. We observe that ProtoCPC enhances the performance of representation learning without the adjustment of hyper-parameters. Also, in Table 7, we demonstrate the effect of prior momentum rate $m_p$ and observe that non-zero $m_p$ results in better representation learning.

| Architecture | Method | *Linear* | *k-NN* |
|---|---|---|---|
| ResNet-50 | DINO | 70.9 | 62.3 |
| | ProtoCPC | **71.8(+0.9)** | **63.7(+1.4)** |
| DeiT-S/16 | DINO | 74.0 | 69.3 |
| | ProtoCPC | **75.0(+1.0)** | **70.6(+1.3)** |

Table 6: Results of representation learning trained by DINO (Caron et al., 2021) and ProtoCPC (ours). All experiments are run by ours with 100 epochs.

| $m_p$ | 0.0 | 0.9 | 0.99 |
|---|---|---|---|
| *Linear* | 73.4 | **75.0** | 74.5 |
| *k-NN* | 69.6 | **70.6** | 70.5 |

Table 7: Ablations on ProtoCPC loss with prior momentum rate $m_p$ on DeiT-S/16 trained for 100 epochs.

### 4.4 ABLATION STUDY

**Cross-entropy v.s. ProtoCPC and Softmax v.s. Sinkhorn-Knopp** The original KD is implemented by cross-entropy loss between the softmax output of the teacher and student network. Our method uses ProtoCPC loss and SK operator for teacher network. We provide ablation studies on the choice of the loss function and probability assignment operator on the subset of supervised model compression tasks. In Table 8, we observe that using the SK operator boosts the performance of KD, and using ProtoCPC loss further enhances the performance.

**Teacher's prototypes for self-supervised model compression** For prototypical self-supervised teachers, one can use the pre-trained prototypes for distillation. We present an ablation between using pre-trained prototypes and copying student's prototypes for teachers. Table 9 show the results on distillation from SwAV ResNet-50 and DINO ResNet-50 teachers. We observe that when distilling from SwAV teacher, using pre-trained prototypes performs better, while distilling from DINO teacher, copying from student's prototypes performs better. We suspect that as SwAV used single prototypes throughout pre-training, the pre-trained prototypes contain representational knowledge. On the other hand, the pre-trained prototypes of DINO lack such representational knowledge.

| Loss | $p_t$ | WRN-40-2 WRN-16-2 | resnet110 resnet20 | resnet110 resnet32 | resnet32x4 resnet8x4 | vgg13 vgg8 |
|---|---|---|---|---|---|---|
| CE | SM | 74.92 | 70.67 | 73.08 | 73.33 | 72.98 |
| | SK | 75.20 | 71.06 | 72.97 | 73.78 | 73.31 |
| ProtoCPC | SM | 75.36 | **71.08** | 73.38 | 74.84 | 73.66 |
| | SK | **75.79** | 71.04 | **73.55** | **75.02** | **73.79** |

Table 8: Ablation study of losses and assignments of $p_t$ on supervised model compression on CIFAR-100. We compare ProtoCPC and cross-entropy (CE), and Sinkhorn-Knopp (SK) operator and softmax (SM) operator. Average over 5 runs.

| Teacher | Method | *Linear* | *k-NN* |
|---|---|---|---|
| SwAV | New P | 60.8 | 54.5 |
| | Old P | **63.1** | **57.7** |
| DINO | New P | **63.5** | **60.2** |
| | Old P | 60.3 | 56.6 |

Table 9: Ablation of setting teacher's prototypes for self-supervised model compression on ResNet-18. New P denotes copying student's prototype and Old P denotes using pre-trained prototypes.

## 5 CONCLUSION

In this paper, we propose prototypical contrastive predictive coding, a simple yet effective method for the distillation of a network by combining the prototypical method and contrastive learning. Our experiments show the effectiveness of our objective on various applications such as supervised/ self-supervised model compression and self-supervised learning by self-distillation.

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

# A  DETAILS IN EXPERIMENTS

## A.1  PSEUDOCODE FOR PROTOCPC LOSS

The PyTorch style pseudo-code for our ProtoCPC is demonstrated in Algorithm 1.

---

**Algorithm 1** ProtoCPC loss PyTorch-style pseudocode.

---

```
# tps, tpt: student and teacher temperatures
# m: prior momentum rate
prior = torch.ones(1, K) # initialize prior with uniform
def ProtoCPC(zt, zs):
    zt = zt.detach()
    pt = SK(zt / tpt)
    zs = zs / tps

    prior = m*prior + (1-m)* K * torch.mean(pt, dim=0) # sum of prior is always K

    loss_align = -torch.sum(pt * zs, dim=1)
    loss_unif = torch.logsumexp(zs + torch.log(prior), 1)
    loss = loss_align + loss_unif
    return loss.mean()
```

---

## A.2  SUPERVISED MODEL COMPRESSION

Many of followings are borrowed from CRD (Tian et al., 2019).

### A.2.1  OTHER METHODS

We compare to the following other state-of-the-art methods from the literature:

1. Knowledge Distillation (KD) (Hinton et al., 2015)
2. Fitnets: Hints for thin deep nets (Romero et al., 2014)
3. Attention Transfer (AT) (Zagoruyko & Komodakis, 2016a)
4. Similarity-Preserving Knowledge Distillation (SP) (Tung & Mori, 2019);
5. Correlation Congruence (CC) (Peng et al., 2019)
6. Variational information distillation for knowledge transfer (VID) (Ahn et al., 2019)
7. Relational Knowledge Distillation (RKD) (Park et al., 2019)
8. Learning deep representations with probabilistic knowledge transfer (PKT) (Passalis & Tefas, 2018)
9. Knowledge transfer via distillation of activation boundaries formed by hidden neurons (AB) (Heo et al., 2019)
10. Paraphrasing complex network: Network compression via factor transfer (FT) (Kim et al., 2018)
11. A gift from knowledge distillation: Fast optimization, network minimization and transfer learning (FSP) (Yim et al., 2017)
12. Like what you like: Knowledge distill via neuron selectivity transfer (NST) (Huang & Wang, 2017)
13. Contrastive representation distillation (CRD) (Tian et al., 2019)

### A.2.2  NETWORK ARCHITECTURES

- Wide Residual Network (WRN) (Zagoruyko & Komodakis, 2016b). WRN-d-w represnets wide resnet with depth $d$ and width factor $w$.
- resnet (He et al., 2016). We use resnet-d to represent **cifar**-style resnet with 3 groups of basic blocks, each with $16$, $32$, and $64$ channels respectively. In our experiments, resnet8 x4 and resnet32 x4 indicate a 4 times wider network (namely, with $64$, $128$, and $256$ channels for each of the block)

- ResNet (He et al., 2016). ResNet-d represents **ImageNet**-style ResNet with Bottleneck blocks and more channels.
- MobileNetV2 Sandler et al. (2018). In our experiments, we use a width multiplier of $0.5$.
- vgg (Simonyan & Zisserman, 2014). the vgg net used in our experiments are adapted from its original ImageNet counterpart.
- ShuffleNetV1 (Zhang et al., 2018), ShuffleNetV2 (Ma et al., 2018). ShuffleNets are proposed for efficient training and we adapt them to input of size 32x32.

### A.2.3 IMPLEMENTATION DETAILS

All methods evaluated in our experiments use SGD.

- For CIFAR-100, we initialize the learning rate as 0.05, and decay it by 0.1 every 30 epochs after the first 150 epochs until the last 240 epoch. For MobileNetV2, ShuffleNetV1 and ShuffleNetV2, we use a learning rate of 0.01 as this learning rate is optimal for these models in a grid search, while 0.05 is optimal for other models.
- For ImageNet, we follow the standard PyTorch practice but train for 10 more epochs. Batch size is 64 for CIFAR-100 or 256 for ImageNet.

The student is trained by a combination of cross-entropy classification objective and a knowledge distillation objective, shown as follows:

$$\mathcal{L} = \alpha \mathcal{L}_{\text{sup}} + \beta \mathcal{L}_{\text{distill}} \tag{18}$$

For the weight balance factor $\beta$, we directly use the optimal value from the original paper if it is specified, or do a grid search with teacher WRN-40-2 and student WRN-16-2. This results in the following list of $\beta$ used for different objectives:

1. KD (Hinton et al., 2015): $\alpha = 0.1, \beta = 0.9$ and $T = 4$
2. Fitnets (Romero et al., 2014): $\alpha = 1, \beta = 100$
3. AT (Zagoruyko & Komodakis, 2016a): $\alpha = 1, \beta = 1000$
4. SP (Tung & Mori, 2019): $\alpha = 1, \beta = 3000$
5. CC (Peng et al., 2019): $\alpha = 1, \beta = 0.02$
6. VID (Ahn et al., 2019): $\alpha = 1, \beta = 1$
7. RKD (Park et al., 2019): $\alpha = 1, \beta_1 = 25$ for distance and $\beta_2 = 50$ for angle. For this loss, we combine both term following the original paper.
8. PKT (Passalis & Tefas, 2018): $\alpha = 1, \beta = 30000$
9. AB (Heo et al., 2019): $\beta = 0$, distillation happens in a separate pre-training stage where only distillation objective applies.
10. FT (Kim et al., 2018): $\alpha = 1, \beta = 500$
11. FSP (Yim et al., 2017): $\beta = 0$, distillation happens in a separate pre-training stage where only distillation objective applies.
12. NST (Huang & Wang, 2017): $\alpha = 1, \beta = 50$
13. CRD (Tian et al., 2019): $\alpha = 1, \beta = 0.8$

Similar to KD, our ProtoCPC used $\alpha = 1, \beta = 1.75$ and temperature $T = 4$. For ablation study in Table 8, we used $\alpha = 0.1, \beta = 0.9$ for CE losses and $\alpha = 1, \beta = 1.75$ for ProtoCPC losses.

### A.3 SELF-SUPERVISED MODEL COMPRESSION

### A.3.1 PRE-TRAINED TEACHERS

- MoCo-v2 (Chen et al., 2020b) uses InfoNCE loss and momentum encoder. Since InfoNCE requires large negatives, they pertain a large queue which updates by first-in-first-out rule with features of momentum encoder. We used the MoCo-v2 ResNet-50 (He et al., 2016) trained for 800 epochs.

- SwAV (Caron et al., 2020) is a prototypical method that generates probability by adopting prototypes. They generate probability of a given feature by computing similarity with respect to prototypes and minimize the cross-entropy between probability outputs of two different views of an image. They used Sinkhorn-Knopp iteration for probability output. Also, they first proposed multi-crops strategy, which additionally use small crops of an image to expedite the training. We used the SwAV ResNet-50 trained for 800 epochs with multi-crops applied.

- DINO (Caron et al., 2021) is a prototypical method which uses momentum encoder. The training progress is similar to SwAV except that they use momentum encoder on the prototypes and use online centering before the softmax operator. They showed that their method is effective in training vision transformer as well as convnet such as ResNet-50. We used the DINO ResNet-50 and DeiT-S/16 (Dosovitskiy et al., 2020) trained for 800 epochs with multi-crops applied.

We archive the checkpoints of teacher models from the author's original implementation.

For each method, we also conduct self-supervised learning on ResNet-18 for fair comparison. We used same hyper-parameters that were used to train ResNet-50 except that we trained for 100 epochs. In Table 4, we report the results of self-supervised learning on ResNet-18.

### A.3.2 NETWORK ARCHITECTURES

We set the projection heads of student network to be same as the teacher network. When teacher is DINO DeiT, the teacher network do not contain batch normalization, but we add batch normalization to projection heads when training ResNet-18 student. Remark that the projection heads of MoCo and SwAV have output dimension of 128, and the projection head of DINO has output of 256. Then we set the number of prototypes to be $K = 65536$ throughout the experiments. For SwAV, since we use pre-trained prototypes, the number of prototypes is 3000. Every features are normalized before computation with prototypes, and prototypes are normalized during the training.

### A.3.3 TRAINING HYPERPARAMETERS

For probability of teacher, we use SK operator with 3 steps of iteration and $\tau_t = 0.04$. For probability of student, we set $\tau_s = 0.1$. The prior momentum for ProtoCPC loss is 0.9. We use SGD optimizer with batch size 512 and weight decay is 1e-4. The learning rate is 0.6 and is decayed by cosine learning rate schedule to 1e-6.

### A.3.4 EVALUATION

For evaluation, we use both linear evaluation protocol and $k$-nearest neighbor classification. For linear evaluation protocol, we freeze the trained weight and train a linear classifier at the top of the frozen feature. We train with SGD optimizer with batch size 256 and use learning rate of 0.3 with 100 epochs. We use cosine learning rate decay schedule and we don't use weight decay. For $k$-NN classification, we follow weighted $k$-NN with $\tau = 0.07$ as done in (Wu et al., 2018).

### A.4 SELF-DISTILLATION

Most of our implementation is borrowed from DINO (Caron et al., 2021). For self-distillation experiments, we only changed cross-entropy loss to ProtoCPC loss. Detailed hyper-parameters are following:

- ResNet-50: we used 2 layers MLP projection head with output dimension 65536. We trained for 100 epochs with 2x224 + 6x96 multi-crops, batch size 512, learning rate 0.6 with cosine annealing schedule, weight decay 1e-4, teacher temperature $\tau_t = 0.04$, student temperature $\tau_s = 0.1$, prior momentum $m_p$=0.9.

- DeiT-S/16: we used 3 layers MLP projection head with output dimension 65536. We trained for 100 epochs with 2x224 + 8x96 multi-crops, batch size 512, learning rate 0.001 with cosine annealing schedule, weight decay 1e-4, teacher temperature $\tau_t = 0.04$, student temperature $\tau_s = 0.1$, prior momentum $m_p$=0.9.

