# OpenReview forum: "Prototypical Contrastive Predictive Coding"
_ICLR.cc/2022/Conference — ICLR 2022 Poster_

### Official Review · Reviewer_dY6T · 2021-10-29

**Correctness:** 3
**Technical Novelty And Significance:** 3
**Empirical Novelty And Significance:** 3
**Recommendation:** 6
**Confidence:** 3

**Main Review:**

As for citation, the original method of KD was introduced in "Model compression, KDD 2006". The authors cite this paper in the Related Work section, but it might be more appropriate to also cite it in the Introduction section.

Please reply to the below questions.

1. Although the paper title called "PROTOTYPICAL CONTRASTIVE PREDICTIVE CODING", the experiments are only conducted on knowledge distillation. Could you provide other evidence of ProtoCPC's advantages (e.g. Mutual Information Estimation and Representation Learning without distillation experiments)? Otherwise, the paper could be called "**Distillation** with PROTOTYPICAL CONTRASTIVE PREDICTIVE CODING".
2. Is it possible to compare CRD with ProtoCPC beyond the Supervised Knowledge Distillation? The authors claim that "Our method outperforms KD as well as CRD
in supervised model compression on CIFAR-100 and ImageNet". From the experiment results, it doesn't necessarily outperform CRD.
3. In **Prior momentum** of 2.1, the authors state that "The prior momentum allows better estimation of prior term regardless of the size of negative sample". However, from the equations (6) (7) & (10), the sampled N is still required in the prior. What is the actual reason that we can ignore a large number of negative samples?
4. Furthermore, as the authors claim their ProtoCPC "does not require massive negative samples" when performing contrastive learning, will this method boost the training efficiency?

Minor comment:

- In Table 3 description, the references of SP and CC are missing.

**Summary Of The Paper:**

This paper proposes to combine knowledge distillation (KD) and contrastive learning for distillation tasks. Concretely, it models the critic of a contrastive objective by the prototypical probabilistic discrepancy between two features. The authors then carry out extensive experiments on supervised model compression, self-supervised model compression, and self-supervised learning through self-distillation. The empirical results show that the proposed method outperforms other strong baselines.

**Summary Of The Review:**

This paper proposes to combine knowledge distillation and contrastive learning for distillation tasks, which is also well-motivated. Extensive experiments on benchmarks validate its effectiveness. It's a nice paper with solid, but somewhat incremental, technical contributions.

---

> ### Author Response · Authors · 2021-11-11
> **Response to the reviewer dY6T [1/2]**
>
> We appreciate your valuable review, and we present answers to your questions below.
>
> [Q1] Although the paper title called "PROTOTYPICAL CONTRASTIVE PREDICTIVE CODING", the experiments are only conducted on knowledge distillation. Could you provide other evidence of ProtoCPC's advantages (e.g. Mutual Information Estimation and Representation Learning without distillation experiments)? Otherwise, the paper could be called "Distillation with PROTOTYPICAL CONTRASTIVE PREDICTIVE CODING".
>
> [A1] As our ProtoCPC objective is proven to be a lower bound to the mutual information, one can expect the effectiveness of the ProtoCPC objective in mutual information estimation or representation learning tasks other than distillation. However, since we only focus on the distillation tasks and we thereby decided to modify the title of the paper to "Representation Distillation by Prototypical Contrastive Predictive Coding".
>
> [Q2] Is it possible to compare CRD with ProtoCPC beyond the Supervised Knowledge Distillation? The authors claim that "Our method outperforms KD as well as CRD in supervised model compression on CIFAR-100 and ImageNet". From the experiment results, it doesn't necessarily outperform CRD.
>
> [A2] As for supervised model compression tasks on CIFAR-100, ProtoCPC outperforms CRD in 8 out of 13 baselines, and on ImageNet, ProtoCPC outperforms CRD. However, we agree that ProtoCPC does not overwhelm CRD, so we revised the statement to be 'partially outperform CRD'
>
> Furthermore, we want to emphasize that ProtoCPC has its advantage in simpler implementation and better computational efficiency. While CRD requires a large memory bank of negative samples (4096 for CIFAR-100 and 65536 for ImageNet), and those negative samples are sampled concerning class information with alias method~\cite{crd}. On the other hand, ProtoCPC is computed in a batch-wise manner (which we used 64 for CIFAR-100 experiments and 256 for an ImageNet experiment) and didn't require any negative sampling tactics or memory bank. Furthermore, while CRD requires additional training parameters to implement the critic, ProtoCPC does not need auxiliary parameters as it reuses the last linear layer for prototypes. In terms of training time and GPU memory consumption, CRD takes 32 sec/epoch of training time and uses 2.6GFLOPS, and our ProtoCPC takes 21sec/epochs of training time and uses 1.5GFLOPS, which shows the efficiency of our method.

---

> ### Author Response · Authors · 2021-11-11
> **Response to the reviewer dY6T [2/2]**
>
> [Q3] In Prior momentum of 2.1, the authors state that "The prior momentum allows better estimation of prior term regardless of the size of negative sample". However, from the equations (6) (7) \& (10), the sampled N is still required in the prior. What is the actual reason that we can ignore a large number of negative samples?
>
> [A3] Through our derivation on ProtoCPC objective in Eq. (7), the $q^{(k)}$ reflects the computation with negative samples. Besides, we provide a simple analysis on the ProtoCPC objective that the prototypes act as a mean of clusters and the $q^{(k)}$ acts as a prior of $k$-th cluster. Then since the $q^{(k)}$ is computed by the mean of $N$ samples, if $N$ is sufficiently large, the estimated $q^{(k)}$ reflects the whole prior over the $K$ prototypes. However, since using large $N$ is computationally burdensome, we use small $N$ and use moving average to reflect the whole trend of prior over $K$ clusters. We empirically show that when using a small batch size (of 512), using momentum averaging on prior results in better performance (Table 7).
>
>
> [Q4] Furthermore, as the authors claim their ProtoCPC "does not require massive negative samples" when performing contrastive learning, will this method boost the training efficiency?
>
> [A4] Since our method is based on a contrastive learning objective, it requires sufficiently large negative samples for computing the loss. Many contrastive learning methods such as SimCLR [1], BYOL [2], SwAV [3] requires large batch size (such as 4096) to fulfill such conditions, or MoCo [4] used large memory buffers (size of 65536). On the other hand, our method can achieve strong performance with only a small batch size of 512 in representation learning. Furthermore, SEED [6] used memory buffers of size 65536, but again, our ProtoCPC based self-supervised representation distillation only used a batch size of 512 and outperforms SEED in distillation from various self-supervised models.
>
> References
>
> [1] Chen, Ting, et al. "A simple framework for contrastive learning of visual representations." International conference on machine learning. PMLR, 2020.
>
> [2] Grill, Jean-Bastien, et al. "Bootstrap your own latent: A new approach to self-supervised learning." arXiv preprint arXiv:2006.07733 (2020).
>
> [3] Caron, Mathilde, et al. "Unsupervised learning of visual features by contrasting cluster assignments." arXiv preprint arXiv:2006.09882 (2020).
>
> [4]  He, Kaiming, et al. "Momentum contrast for unsupervised visual representation learning." Proceedings of the IEEE/CVF Conference on Computer Vision and Pattern Recognition. 2020.
>
> [5] Fang, Zhiyuan, et al. "Seed: Self-supervised distillation for visual representation." arXiv preprint arXiv:2101.04731 (2021).

---

### Official Review · Reviewer_hGXe · 2021-10-31

**Correctness:** 4
**Technical Novelty And Significance:** 3
**Empirical Novelty And Significance:** 3
**Recommendation:** 6
**Confidence:** 3

**Details Of Ethics Concerns:**

I think there are no ethical concerns.

**Main Review:**

Strengths:
This paper is well written and organized.
The proposed method can be integrated into other methods and Experiments are done on different tasks to validate effectiveness.
Weakness:
The main concern is the novelty of this method. It combines the prototypical method and contrastive learning while these two techniques are well researched in the literature. Also, compared with CRD+KD  the improvement is marginal in the knowledge distillation task.

After rebuttal:
I keep my original rate.  I think this paper is well presented with decent results.

**Summary Of The Paper:**

This paper proposes a prototypical contrastive predictive coding by combining the prototypical method and contrastive learning. Experiments are done on applications such as supervised/ self-supervised model compression and self-supervised learning by self-distillation to validate its effectiveness.

**Summary Of The Review:**

This paper is well written and experiments are done thoroughly. The main concern is the novelty as it combines existing techniques without substantial findings.

---

> ### Author Response · Authors · 2021-11-11
> **Response to the reviewer hGXe**
>
> We appreciate your valuable comments. Even though knowledge distillation and contrastive learning is a well-studied topic in the machine learning community, our contributions lay in proposing a new objective called ProtoCPC that combines both of them and efficient implementation of ProtoCPC with application on various distillation tasks such as supervised model compression. Moreover, we proposed a new state-of-the-art method for self-supervised representation distillation, and show efficiency in self-supervised representation learning.

---

### Official Review · Reviewer_tvAr · 2021-11-02

**Correctness:** 4
**Technical Novelty And Significance:** 3
**Empirical Novelty And Significance:** 3
**Recommendation:** 8
**Confidence:** 3

**Main Review:**

Strength:
(i) The proposed method is well described and seems to work as shown by sufficient experiments in various distillation settings.
(ii) Fairly well-written paper and easy to follow various distillation settings.

Weakness:
(i) A detailed justification on why momentum-based estimation helps the ProtoCPC loss is missing?
(ii) Empirically the method seems to work, a critique on why prototypes (clustering) help is missing? Motivation on prototypes would be appreciated.

Questions:
(i) The pseudo-code of the ProtoCPC loss only takes z_t and z_s as input. How are the negative examples provided for momentum estimation?
(ii) Currently, the prototypes are computed using a linear layer. Are more complex networks helpful or deterimental?
(iii) Is the runtime of the method a criterion to evaluate efficient learning?

Typo:
computation between z_s and z_tjS -> computation between z_s and z_tj


**Summary Of The Paper:**

This paper provides a method for combining contrastive learning and clustering (prototypical probabilities) for three knowledge distillation tasks - supervised model compression, self-supervised model compression, and self-supervised learning with self-distillation.
Traditional contrastive learning methods rely on the similarity of feature vectors from the teacher and student networks using cosine similarity, log bilinear model (van den Oord et al), etc. This paper proposes to project the feature vectors into a probability space via linear prototypes (layer) of the same dimension (i.e, cluster the feature vectors). The proposed method overcomes the disadvantage of having large negative examples for contrastive learning by maintaining an EMA of prior that accounts for the negative examples.

The main contribution of this paper is the ProtoCPC loss that combines the advantages of contrastive predictive coding loss using prototypes over the teacher and student representation.

The authors empirically show that the proposed method is useful for various distillation tasks on image classification tasks on CIFAR-100 and Imagenet datasets.

**Summary Of The Review:**

I enjoyed reading the paper and feel it has value to the community. The experimental study justifies the usefulness of the paper.

---

> ### Author Response · Authors · 2021-11-11
> **Response to the reviewer tvAr [1/2]**
>
> We appreciate your fruitful reviews. We present responses to the raised concerns and questions below.
>
> [W1] A detailed justification on why momentum-based estimation helps the ProtoCPC loss is missing?
>
> [A1] Through our derivation on ProtoCPC objective in Eq. (7), the $q^{(k)}$ reflects the computation with negative samples. Besides, we provide a simple analysis on the ProtoCPC objective that the prototypes act as a mean of clusters and the $q^{(k)}$ acts as a prior of $k$-th cluster. Then since the $q^{(k)}$ is computed by the mean of $N$ samples, if $N$ is sufficiently large, the estimated $q^{(k)}$ reflects the whole prior over the $K$ prototypes. However, since using large $N$ is computationally burdensome, we use small $N$ and use moving average to reflect the whole trend of prior over $K$ clusters. We empirically show that when using a small batch size (of 512), using momentum averaging on prior results in better performance (Table 7).
>
> [W2] Empirically the method seems to work, a critique on why prototypes (clustering) help is missing? Motivation on prototypes would be appreciated.
>
> [A2] First, we want to emphasize that many existing studies have empirically shown the effectiveness of using prototypes (or clustering) in learning representation learning [1, 2, 3, 4]. However, those works focus on the clustering methods rather than training objectives.
>
> We presumed that prototypes (or clusters) are variational latent variables $Z$ that acts as a latent class of data distribution $X$. Our ProtoCPC objective models the critic by the cross-entropy $H(P(X_1|Z), P(X_2|Z))$. From a recent study [5], one can show that ProtoCPC objective is a lower bound to the conditional mutual information $I(X_1; X_2 |Z)\leq I(X_1; X_2)$. That mutual information maximization approach is a well-studied approach in representation learning. Moreover, learning prototypes with ProtoCPC objective is thus reducing the gap $I(X_1; X_2) - I(X_1; X_2 | Z)$.
> In sum, using prototypes or clustering has been empirically shown to be effective in representation learning. On the other hand, there were no studies on the training objectives. Therefore, we provide a principled approach by regarding the prototypes as a variational parameter.

---

> ### Author Response · Authors · 2021-11-11
> **Response to the reviewer tvAr [2/2]**
>
> [Q1] The pseudo-code of the ProtoCPC loss only takes $z_t$ and $z_s$ as input. How are the negative examples provided for momentum estimation?
>
> [A1] Remark that $z_t$ and $z_s$ are sizes of $N\times K$, where $N$ is a batch size and $K$ is a number of prototypes. Then we calculate $p_t$ using $z_t$ and the prior is updated by mean of $N$ probabilities in $p_t$.
>
> [Q2] Currently, the prototypes are computed using a linear layer. Are more complex networks helpful or detrimental?
>
> [A2] If we use MLP instead of a linear layer for both teacher and student networks, it is simply a deeper encoder. On the other hand, two papers [4, 6] empirically showed that attaching an additional MLP layer to the student network degrades performance slightly.  Since the prototypes are only for computing $K$ categorical distributions, we believe that using a linear layer suffices.
>
> [Q3] Is the runtime of the method a criterion to evaluate efficient learning?
>
> [A3] For your information, when performing knowledge distillation tasks on CIFAR-100 (teacher network: wide-resnet 40-2, student network: wide-resnet-16-2), CRD takes 32 sec/epoch of training time with 2.6GFLOPS GPU memory consumption, while ProtoCPC takes 21 sec/epoch of training time with 1.46GFLOPS of GPU memory consumption. Therefore, our method enjoys faster training time and lower GPU memory consumption, showing efficiency.
>
> References
>
> [1] Tian, Kai, Shuigeng Zhou, and Jihong Guan. "Deepcluster: A general clustering framework based on deep learning." Joint European Conference on Machine Learning and Knowledge Discovery in Databases. Springer, Cham, 2017.
>
> [2] Asano, Yuki Markus, Christian Rupprecht, and Andrea Vedaldi. "Self-labelling via simultaneous clustering and representation learning." arXiv preprint arXiv:1911.05371 (2019).
>
> [3] Caron, Mathilde, et al. "Unsupervised learning of visual features by contrasting cluster assignments." arXiv preprint arXiv:2006.09882 (2020).
>
> [4] Caron, Mathilde, et al. "Emerging properties in self-supervised vision transformers." arXiv preprint arXiv:2104.14294 (2021).
>
> [5] Tsai, Yao-Hung Hubert, et al. "Conditional Contrastive Learning: Removing Undesirable Information in Self-Supervised Representations." arXiv preprint arXiv:2106.02866 (2021).
>
> [6] Chen, Xinlei, and Kaiming He. "Exploring simple siamese representation learning." Proceedings of the IEEE/CVF Conference on Computer Vision and Pattern Recognition. 2021.

---

### Official Review · Reviewer_EXkK · 2021-11-03

**Correctness:** 4
**Technical Novelty And Significance:** 2
**Empirical Novelty And Significance:** 2
**Recommendation:** 6
**Confidence:** 4

**Main Review:**

I found this to be an interesting paper. The results look decent, the method makes sense, and the derivations appear to be sound. The paper is clearly written and well presented.

There are however, several weaknesses to the paper. First and foremost, almost all the methods appear in prior work, albeit used in slightly different ways. The use of representations of negatives, updated with momentum, is reminiscent of MoCo; in both cases the basic method -- momentum -- and outcome -- smaller batches of negatives -- are the same, even though how the negatives are represented is different. Prior work has used contrastive losses for knowledge distillation (e.g., CRD), and has used prototypes for contrastive learning (e.g., DINO, SwAV). The Sinkhorn-Knopp method was introduced in SeLa. The present paper is novel in how it puts together these pieces and applies them, but the key technical ideas are all already in the literature, as far as I can tell.

Nonetheless, as an engineered system, the paper achieves solid performance. The ablations and comparisons are thorough, and I'm convincined that the ProtoCPC objective does have benefits over the baselines, although the numerical gains are not large.

One point the paper makes is that ProtoCPC avoids the large batches required by some of the competing methods, and that this could have advantages in terms of computational efficiency. I would have liked to see a quantification of this point, in terms of total number of network calls, memory consumption, or wall clock time.

Two minor comments:
1. I'm not sure the name ProtoCPC is well-chosen. Contrastive Predictive Coding involved a "predictive" component in the form of the linear transformation W that predicts z_{t+k} from c_t. I don't really see a "prediction" component in present paper, at least not in the same sense as in CPC.
2. The title also could be improved: there is no mention of distillation in the title yet the rest of the paper is all about applications to distillation.
3. “Our method infers” —> “Our method inherits”?

**Summary Of The Paper:**

This paper studies a new contrastive loss function for knowledge distillation and self-supervised representation learning. The paper demostrates results on transferring knowledge from a teacher to student using both supervised and self-supervised objectives, and on representation learning based on self-distillation. As prior work has already applied contrastive learning to each of these problems the main novelty of the paper is the exact form of the contrastive loss. The loss is computed over projections of the embeddings into a k-dimensional simplex (probability space) where cross-entropy can be used as the similarity function. This allows for encoding "negatives" into a distribution over k prototypes that are updated with EMA over iterations of training rather than requiring a large batch of negatives during each iteration of training. The results appear to slightly outperform SOTA methods.

**Summary Of The Review:**

I think this is a fine paper, well executed, and with decent results. However, I think the novelty is minimal compared to prior work that introduced the key methods that are used, and the empirical gains are also not that large. I therefore think this paper is slightly below the threshold of typical ICLR papers in terms of value to the community, although it is above the threshold in terms of sound science.

---

> ### Author Response · Authors · 2021-11-11
> **Response to the reviewer EXkK**
>
> We sincerely appreciate your valuable comments. We list the response to the concerns below.
>
> [Q1] Regarding the novelty of our work
>
> [A1] Even though the idea of prototypical representation learning is not new, most of the existing methods focus on clustering methods rather than training objectives. We believe that our contribution is proposing the new ProtoCPC objective that enjoys the advantage of contrastive learning. Moreover, we provide an efficient implementation of a ProtoCPC objective (such as prior momentum) and application to various distillation tasks with decent empirical results.
>
> [Q2] Regarding the computational efficiency of our work
>
> [A2] When performing knowledge distillation tasks on CIFAR-100 (teacher network: wide-resnet 40-2, student network: wide-resnet-16-2), CRD takes 32 sec/epoch of training time with 2.6GFLOPS GPU memory consumption, while ProtoCPC takes 21 sec/epoch of training time with 1.46GFLOPS of GPU memory consumption. Therefore, our method enjoys faster training time and lower GPU memory consumption, showing efficiency.
>
> [Q3] Regarding the title of our work
>
> [A3] As reviewer dY6T noticed, since our method focus on distillation tasks, we decided to change the title of our paper to "Representation Distillation by Prototypical Contrastive Predictive Coding".
> While CPC is originated from predicting the future with the latent class [1], many recent works [2,3] interpreted CPC as a low variance estimator of mutual information. Therefore, since our objective is based on CPC, and is also a lower bound estimator of mutual information, we believe ProtoCPC is a proper terminology of our method.
>
> References
>
> [1] Oord, Aaron van den, Yazhe Li, and Oriol Vinyals. "Representation learning with contrastive predictive coding." arXiv preprint arXiv:1807.03748 (2018).
>
> [2] Poole, Ben, et al. "On variational bounds of mutual information." International Conference on Machine Learning. PMLR, 2019.
>
> [3] Song, Jiaming, and Stefano Ermon. "Multi-label contrastive predictive coding." arXiv preprint arXiv:2007.09852 (2020).

---

> > ### Comment · Reviewer_EXkK · 2021-11-29
> > **thanks for the response, upgraded rating**
> >
> > Thanks for your response to my questions and those of the other reviewers.
> >
> > I think the new title is better, and it would also be great to add the timing comparison between CRD and ProtoCPC, mentioned in your response. I still think the "predictive" part of CPC is missing -- I would say this is a purely "contrastive" approach -- but it's not a big deal, definitely people have already been using the term "CPC" in the way you are using it.
> >
> > Upon reflection of the paper, the other reviews, and the responses, I'm feeling more positive about the novelty. The I_{ProtoCPC} objective is indeed new, as far as I can tell, even if it does build on existing ideas. And it's nice and clean.
> >
> > Therefore, I am upgrading my rating to 6, and I will be happy to see this at the conference.

---

> > > ### Author Response · Authors · 2021-11-30
> > > **Thanks for your positive response!**
> > >
> > > It is nice that you appreciate our new paper title and responses to the reviews. Also, we sincerely appreciate that you understand the novelty of our work and raising the score.
> > >
> > > If there is any further questions we'll be pleased to answer. Thank you!

---

### Author Response · Authors · 2021-11-11
**Response to the reviewers**

We sincerely appreciate all the valuable comments from the reviewers. Here we list some major changes in our paper.

- As reviewer EXkK and dY6T noticed, we changed the title of the paper to 'Representation Distillation by Prototypical Contrastive Predictive Coding' since our method is for distillation tasks.
- Some references and minor statements are revised.

Again, thank you for the fruitful reviews and we appreciate any suggestions on our work!

---

### Decision · Program_Chairs · 2022-01-20

**Decision:**

Accept (Poster)

**Comment:**

This paper proposes a prototypical contrastive predictive coding by combining the prototypical method and contrastive learning, and presents its efficient implementation for three distillation tasks: supervised model compression, self-supervised model compression, and self-supervised learning via self-distillation. The paper is well-written, and the effectiveness of the proposed method is validated through extensive experiments.  Reviewers generally agree the paper has clear merits despite some weaknesses for improvement. Overall, I would like to recommend it for acceptance and encourage authors to incorporate all the review comments and suggestions in the final version.